# Influence of Nutritional Stress on Female Allocation and Somatic Growth in the Simultaneously Hermaphroditic Polychaete Worm *Ophryotrocha diadema*

**DOI:** 10.3390/biology12060859

**Published:** 2023-06-14

**Authors:** Dáša Schleicherová, Marino Prearo, Alfredo Santovito

**Affiliations:** 1Department of Life Sciences and Systems Biology, University of Turin, Via Accademia Albertina 13, 10124 Torino, Italy; alfredo.santovito@unito.it; 2IZS PLV, Istituto Zooprofilattico Sperimentale del Piemonte, Via Bologna 148, 10154 Torino, Italy; marino.prearo@izsto.it

**Keywords:** egg cocoons, marine polychaetes, food deprivation, female investment, life–history traits

## Abstract

**Simple Summary:**

Simultaneous hermaphrodites mainly adjust their reproductive resources according to mating opportunities. Numbering among other factors in the overall energy budget of hermaphrodites are life–history traits of species, environmental conditions and stressors. All such factors may influence the allocation of reproductive resources in hermaphrodites and the trade-off between sexual functions. Here, we investigated the trade-off between food deprivation and the investment of resources in female allocation and somatic growth in the simultaneous hermaphrodite *Ophryotrocha diadema*, a polychaete worm, by exposing focal individuals to three different nutritional regimes. Our findings show the marked influence of nutritional stress on female allocation and body growth rate.

**Abstract:**

Hermaphrodites are characterized by plastic sex allocation, by which they adjust their allocation of reproductive resources according to mating opportunities. However, since the plasticity of sex allocation is influenced by environmental conditions, it may also be affected by species-specific life–history traits. In this study, we explored the trade-off between nutritional stress due to food deficiency and the investment of resources in female allocation and somatic growth in the simultaneously hermaphroditic polychaete worm, *Ophryotrocha diadema*. To achieve this, we exposed adult individuals to three food supply levels: (1) ad libitum—100% food supply, (2) intense food deficiency—25% food resources, and (3) extreme food deficiency—0% food resources. Our findings show a progressive decrease in female allocation in the numbers of cocoons and eggs and in body growth rate of *O. diadema* individuals as the level of nutritional stress increased.

## 1. Introduction

Simultaneous hermaphrodites invest their reproductive resources in two sexual functions (male and female) in response to changes in mating opportunities via the plastic adaptation of their reproductive resources, in which they increase or reduce their resource allocation to one sex at the expense or in favour of the other [1]. Charnov’s Sex Allocation Theory [1] posits that in large mating groups (i.e., promiscuity) where there are many mating opportunities, simultaneous hermaphrodites are predicted to increase their investment in the male function at the expense of the female function. However, in small mating groups (i.e., monogamy) where mating opportunities are few, hermaphrodites are predicted to decrease their investment in the male function in favour of the female function, since the male function requires resources to produce sperm for fertilizing the eggs of a single partner. Thus, hermaphrodites are able to flexibly adjust their reproductive resources to environmental and social cues and trade-offs between the two sexual functions. These predictions are supported by empirical tests [2,3,4,5,6,7,8,9,10,11,12,13].

The way that hermaphrodites adjust their reproductive resources to mating opportunities varies between species. Experimental studies have revealed that the trade-offs observed between sexual functions often differ from those predicted. For example, species such as polycheate worms [2,7,10,13] or pond snails [14] have a plastic female allocation and a fixed male allocation. Conversely, other species such as leeches [15], shrimps [8], flatworms [16], and barnacles [17] have a plastic male allocation but maintain their fixed female allocation. Such diversity in the patterns of phenotypic plasticity in sex allocation may be caused by the variety of the model species used in experimental testing and of the experimental conditions specific to each species [18].

Sex allocation adjustments in the outcrossing hermaphroditic polychaete worm *Ophryotrocha diadema* differ slightly from Charnov’s theoretical model: in large mating groups (promiscuity), the number of female gametes is significantly lower than that produced by hermaphrodites in small mating groups (monogamy), but the number of sperm is fixed in both mating regimes [2,13]. Differently, resources subtracted from the female function are not directly attributed to the male function in *O. diadema* but rather to expensive behaviours such as direct competition between individuals [4]. In other words, there is a trade-off between female allocation and direct competition, a mating behaviour trait, and not between sexual functions as posited by Charnov’s model.

Moreover, further experimental tests have revealed a trade-off between male allocation and life–history traits such as somatic growth rate and lifespan, in which the cost of male reproduction (fertilization rate) during the protandrous phase is spread over the entire energy budget, resulting in shorter lifespan and lower growth rates in individuals with a higher male expenditure during the protandrous phase (higher fertilization rate during adolescence). In their empirical study, Locher and Baur [19] examined the effect of nutritional stress on mating behaviour and male and female reproductive output in the simultaneously hermaphroditic land snail, *Arianta arbustorum*, which was exposed to three different food regimes (ample, restricted, and extremely restricted food supplies). They found that snails exposed to a restricted food supply produced fewer eggs, whereas nutritional stress did not affect the number of sperm produced. Instead, they observed altered mating behaviour: courtship and copulation were affected by nutritional stress. They hypothesized that two trade-offs were at play in *A. arbustorum* snails, trade-offs that do not directly involve both sexual functions according to Charnov’s model, but rather a trade-off between nutritional stress and female allocation, as well as a trade-off between nutritional stress and mating behaviours.

Here, we explored the trade-off between nutritional stress caused by food deficiency and investment of reproductive resources in female allocation in the simultaneously hermaphroditic polychaete worm *Ophryotrocha diadema*. Nutritional stress was induced via the exposure to three different levels of food supply: (1) ad libitum—100% food resources, (2) intense food deficiency—25% food resources, and (3) extreme food deficiency—0% food resources. Since previous studies showed that the number of sperm is not influenced by social and environmental conditions [2,4,12,13], we estimated the influence of nutritional stress only on female allocation (number of cocoons laid and number of eggs/cocoon). In addition, to determine the influence of nutritional stress on somatic growth rate, we estimated body size by counting the number of setigerous segments at the beginning and end of the 3-week study period and estimated body growth rate from the difference between the final and the initial number of segments. Drawing on the observations of Locher and Baur [19], we hypothesized a progressive decrease in female allocation and somatic growth rate with increases in nutritional stress.

### Study Model

*Ophryotrocha diadema*, a 1 mm long marine polychaete worm, is a simultaneous hermaphrodite with external fertilization. All data on life cycle [20] and mating [20] were obtained via laboratory observations. *O. diadema* inhabits the organic sediments of fouling fauna in Californian harbours and belongs to a marine zooplankton species. Despite living in low-density populations, adults produce a network of mucous trails that can be followed by conspecifics, and the spatial distribution of animals is likely clustered (G. Sella, personal observation). The life cycle consists of a protandrous phase, followed by a simultaneously hermaphroditic phase that starts when individuals are 14 to 15 segments in length [20] and lasts for approximately 30–40 days. Mating is preceded by courtship, during which both partners rub mutually against each other and via pseudocopulation or external fertilization, in which the partners maintain close physical contact before releasing their gametes (as described for *O. gracilis* by Westheide [21]).

The sperm of *O. diadema* are immotile [22]. Despite being simultaneous hermaphrodites with simultaneously mature and viable oocytes and sperm, they never lay and do not self-fertilize if isolated. Instead, monogamous partners regularly alternate sex roles every 2 days via the reciprocal and conditional exchange of a transparent cocoon containing 20–25 eggs. This reproductive strategy is termed *egg trading* [23]. Since the cocoons are transparent, egg development can be followed under a stereomicroscope. Nine days after laying, offspring hatch as small four-segment individuals, soon ready to produce their first sperm. When the worms reach a length of 14 segments, they become simultaneous hermaphrodites: they produce sperm in the fourth and the fifth segment and eggs in the following segments [20]. The species exhibits two different phenotypes: yellow (YY, Yy) and white (yy) phenotypes. Their maximum body length is 20–21 segments and their life span is approximatively 90 days.

## 2. Materials and Methods

### 2.1. Experimental Set Up

The experiment was carried out in 10 mL glass bowls filled with artificial sea water (salinity 35‰), placed in closed boxes to reduce evaporation, and kept in a thermostatic cabinet at 20 °C. Temperature and dissolved oxygen were kept constant during the study period. The aim of the experiment was to test the effects of nutritional stress on female allocation and body growth rate. To promote the *egg trading* reproductive strategy of adult virgin hermaphrodites of the same age, 32 *O. diadema* parent pairs (PP) with no siblings or offspring (F1) were randomly divided into three experimental groups (Table 1).

Experimental group A (controls; nutritional stress—ABSENT). Two adult individuals of *O. diadema*/bowl: 1 Yy with yellow phenotype and 1 yy with white phenotype were fed ad libitum with 400 µL of aqueous suspension of minced parboiled spinach, at a level of 100% feeding supply. This amount is usually administered during the rearing of *O. diadema* pairs in a laboratory.

Experimental group B (nutritional stress—INTENSE). Two adult individuals of *O. diadema*/bowl: 1 Yy with yellow phenotype and 1 yy with white phenotype were fed with 100 µL of an aqueous suspension of minced parboiled spinach, at a level of 25% feeding supply.

Experimental group C (nutritional stress—EXTREME). Two adult individuals of *O. diadema*/bowl: 1 Yy with yellow phenotype and 1 yy with white phenotype were not fed at all, a level of 0% feeding supply.

The total number of replicates was 24 for each group, and the total number of individuals was 144 (72 with yellow phenotype *O. diadema* Yy and 72 with white phenotype *O. diadema* yy). Female reproductive parameters were the number of cocoons laid and the number of eggs/cocoon/individual and life–history characteristics, as well as somatic growth rate, defined as the number of setigerous segments (the difference between the number of initial setigerous segments and the number of final setigerous segments at the end of the study period). All reproductive parameters were estimated twice a week during the 3-week study. In order to estimate female reproductive output and body size, the parameters were only measured in focal individuals (*O. diadema* Yy with yellow phenotype).

### 2.2. Statistical Analysis

Statistical analysis was performed using IBM-SPSS 27.0 for Windows (IBM-SPSS, Armonk, NY, USA). Descriptive statistics are reported as means ± standard deviation (SD). The distribution of number of cocoons, eggs/individual, and growth rate was tested for normality using the Kolmogorov–Smirnov one-sample test and for homogeneity of variances using Levene’s test (*p* > 0.05). In the first analysis, we used ANOVA to check whether *O. diadema* focal individuals responded differentially to treatment. The number of cocoons, the number of eggs per individual, and the growth rate were the dependent variables, and treatment (level of nutritional stress) was a fixed factor. Moreover, we checked for differences in female reproductive output and growth rate between the three treatments using Tukey post hoc tests. The level of significance was set at *p* < 0.05 and the probabilities were two-tailed. Data analysis was performed only on focal worms that were alive at the end of the experiment.

## 3. Results

### 3.1. Influence of Nutritional Stress on Female Allocation in O. diadema Focals

Group A (Controls) experienced no nutritional stress because of ad libitum access to food (100% nutritional regime). The mean number of cocoons laid by focals was 5.29 ± 1.33. In contrast, in groups B (25% nutritional regime) and C (0% nutritional regime), the mean number of cocoons was reduced (3.71 ± 1.55 in group B and 0.75 ± 0.79 in group C; ANOVA, descriptive statistics). There was a stark difference in the mean number of eggs/cocoon between the three groups: 111.62 ± 36.84 in group A, 33.83 ± 14.74 in group B, and 4.04 ± 4.60 in group C (ANOVA, descriptive statistics).

ANOVA showed that female allocation, as measured by the number of cocoons laid and the number of eggs/cocoon, was influenced by nutritional stress (mean number of cocoons, df 2, F_2,127_ 79.702, ANOVA, *p* < 0.001; mean number of eggs/cocoon, df 2, F_2,74053_ 139.251, ANOVA, *p* < 0.001) (Figure 1 and Figure 2).

The Tukey post hoc test revelaed a significant effect of nutritional stress on female allocation (mean number of cocoons and mean number of eggs/cocoon) between the three groups, with marked differences between groups A and B, B and C, and A and C (Tukey post hoc test, *p* < 0.001).

### 3.2. Influence of Nutritional Stress on Growth Rate in O. diadema Focals

The mean growth rate in group A was 3.29 ± 1.46 setigerous segments, while the growth rate in groups B and C was reduced (2.17 ± 1.31 and 1.46 ± 1.82 setigerous segments in group B and group C, respectively (ANOVA, descriptive statistics). The growth rate was influenced by nutritional stress (mean growth rate, df 2, F_2,20_ 8618, ANOVA, *p* < 0.001 (Figure 3).

In addition, we checked differences in growth rate between groups (A–B, B–C, A–C) using Tukey post hoc tests, which showed a significant and a highly significant effect of nutritional stress on growth rates between groups A and B and groups A and C, respectively, but not between groups B and C (Tukey post hoc test, A and B, *p* = 0.036; A and C, *p* < 0.001; B and C, *p* > 0.05).

## 4. Discussion

Charnov’s sex allocation theory (1982) posits that, because the amount of reproductive resources is limited, a higher investment in one sexual function leads to a lower investment in the other sexual function in a trade-off between the two. However, the few studies that tested Charnov’s sex allocation theory have ambiguous results, since the trade-offs observed between sexual functions often differed from those predicted (for example, in the snail, *Arianta arbustorum* [19]; colonial bryozoan, *Celleporella hyalina* [24]; free living flatworm, *Macrostomum lignano* [16]; snail, *Lymnea stagnalis* [14]; and marine polychaete worms, *Ophryotrocha diadema*, *O. adherens*, and *O. gracilis* [2,7,12,13]). These studies revealed the plasticity of one sexual function, while the other remains fixed.

Among the factors involved in the overall energy budget of hermaphrodites are the life–history traits of species (mating behaviour such as mate guarding, mate searching, courtship, fertilization rate, food competition, food searching) and environmental conditions and stressors (physical obstruction, temperature shock, physical damage, food deprivation). All of these factors may influence the allocation of reproductive resources in hermaphrodites, as well as the trade-off between sexual functions.

Two factors (the fertilization rate during the protandrous phase and direct competition between individuals) were tested in *O. diadema* to understand their implications for the overall energy budget, allocation of reproductive resources, and the trade-offs between sexual functions [4]. A previous study [4] reported that the cost of male reproduction during the protandrous phase is spread over the entire energy budget, resulting in shortened lifespans and lower somatic growth rates in individuals with greater male expenditure. Moreover, the study also showed that, in large mating groups (promiscuity) with many reproductive competitors present, the resources subtracted from the female function are not directly allocated to the male function but rather to interindividual conflicts (direct competition), with relevant costs in the entire energy budget of individuals. Another study [4] found three trade-offs between one sexual function and life–history traits: a trade-off between male function and somatic growth rate, a trade-off between male function and lifespan, and a trade-off between female function and direct competition. Experimental evidence for stress-induced maleness in the marine bryozoan *Celleporella hyaline* was reported in [24]. Environmental stressors, naturally encountered by *C. hyalina* and simulated in a laboratory included physical obstruction (physical impediment to growth), reduced food supply, temperature shock or desiccation and physical damage. The effects of such stressors on the sex allocation of *C. hyalina* colonies revealed an increased proportional allocation to males in response to environmental stressors, mainly physical obstruction, temperature shock, desiccation, physical damage, and food deprivation. According to one study [25], environmental stress promotes maleness in *C. hyalina* and leads to a trade-off between male function and environmental stressors.

Locher and Baur [19] studied the effects of food deprivation in the simultaneous hermaphroditic gastropod *Arianta arbustorum* and found a significant effect of food deprivation (nutritional stress) on courtship mating and copulation behaviour, as well as the allocation of reproductive resources to the female function. The study showed that changes in sex-specific reproductive allocation were caused by nutritional stress and provided further evidence for the trade-off between one sexual function, female function in this case, and environmental conditions, such as food deprivation or nutritional stress.

In the present study, we aimed to determine whether a trade-offs exist between nutritional stress and the investment of reproductive resources in female function (number of cocoons/focal and number of eggs/cocoon/focal), as well as between nutritional stress and the somatic growth rates of focal individuals (number of setigerous segments) in the simultaneously hermaphroditic polychaete worm, *O. diadema*. Drawing on a previous study [19] and a food deprivation experimental test described in [24], our hypothesis was that there would be a progressive decrease in female allocation and somatic growth rates with an increase in nutritional stress. Our findings show that the female allocation and somatic growth rate of *O. diadema* focals were strongly influenced by nutritional stress.

In detail, female allocation, as defined by the number of cocoons laid by focals and the number of eggs/cocoon/focal, was strongly influenced by nutritional stress. The female reproductive output in group A was comparatively higher because the focals had ad libitum access to food and were not exposed to nutritional stress. Differently, exposure to nutritional stress was intense and female reproductive investment significantly decreased in group B as well as in group C exposed to extreme nutritional stress.

The differences in female allocation between the three groups (A and B, A and C and B and C) were highly significant. Similarly, the somatic growth rate of *O. diadema* focals was also significantly influenced by nutritional stress. Compared to group A, the somatic growth rates of focals were slower in groups B and C, exposed to intense and extreme nutritional stress, respectively. There was a significant difference in somatic growth between groups A and B, a highly significant difference between groups A and C, but no significant differences in somatic growth between groups B and C, in which the somatic growth rate was similar. Our data show that nutritional stress plays a fundamental role in the allocation of resources to reproduction and somatic growth of *O. diadema* focals. Our observations are shared by [19,24] and provide evidence for two trade-offs in *O. diadema*: a trade-off between nutritional stress and female function and a trade-off between nutritional stress and somatic growth.

## 5. Conclusions

Global warming has thus far changed the phenology, abundance and distribution of many taxa in marine ecosystems and affects all living taxa on earth [25]. However, a plausible reduction in phytoplankton biomass richness due to the temperature increase is also predicted [25,26]. Our study supports the importance of the availability of food resources in *O. diadema*. As shown in our results, the decrease in trophic richness could have a strong impact on the reproductive output and inevitably on the population growth rate of this marine polychaete worm. Consequently, such an impact on reproduction can potentially lead to repercussions at the level of the food chain, biodiversity loss and even to extinction events.

Our study findings underscore the importance of the availability of food resources in *O. diadema*. Food deprivation may be variable, occasional or frequent in natural environments depending on environmental conditions. This is a limitation of our study because we can only simulate food deprivation in the laboratory. Moreover, replicating all possible conditions in the laboratory such as those that can occur in nature is difficult.

Abiotic factors, such as increased temperature and acidification, are known to have a significant impact on the biomass of phytoplankton and zooplankton communities and their general survival, behaviour, reproduction, and overall population dynamics. For example, evidence suggests that marine zooplankton experience above-subsistence food levels caused by a reduction in phytoplankton biomass richness associated with the rise in marine water temperature [26]. Similarly, a reduction in trophic richness could have a strong impact on reproductive output and on the population growth rate of this marine polychaete worm. Considering that zooplankton provides the main trophic connection between primary producers (phytoplankton) and final consumers, any biotic or abiotic factor that interferes with their population dynamics may have repercussions on the food chain and biodiversity.

## Figures and Tables

**Figure 1 biology-12-00859-f001:**
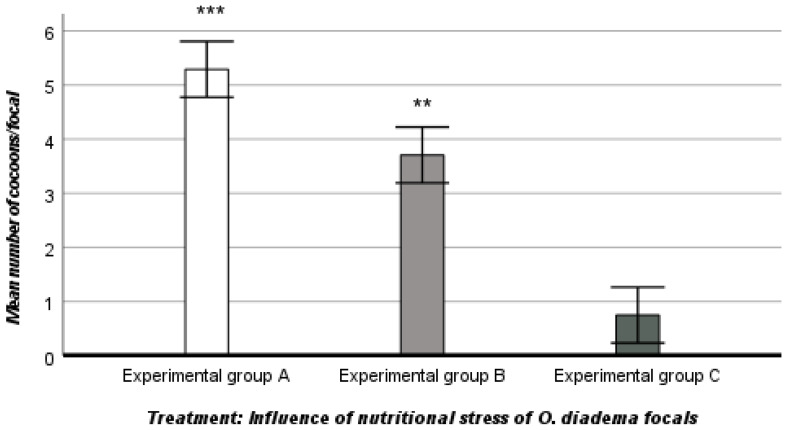
Influence of nutritional stress on female allocation (mean number of cocoons/focal) in the three groups (A, B, C). *** *p* < 0.001, significantly higher than groups B and C (Tukey post hoc test). ** *p* < 0.001, significantly higher than group C (Tukey post hoc test).

**Figure 2 biology-12-00859-f002:**
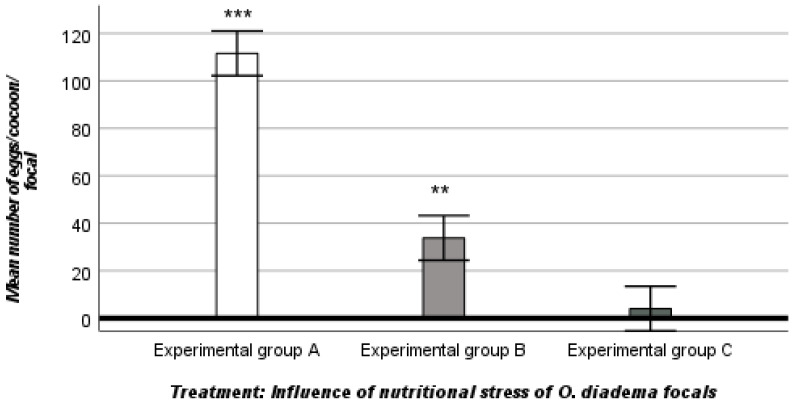
Influence of nutritional stress on female allocation (mean number of eggs/cocoon focal) in the three groups (A, B, C). *** *p* < 0.001: significantly higher than groups B and C (Tukey post hoc test). ** *p* < 0.001: significantly higher than group C (Tukey post hoc test).

**Figure 3 biology-12-00859-f003:**
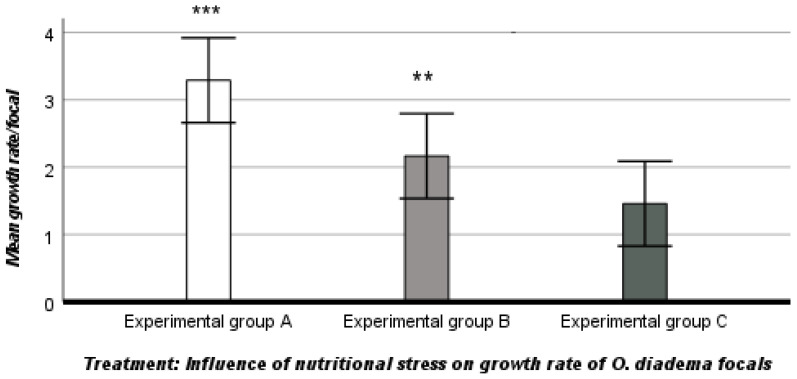
Influence of nutritional stress on growth rate in the three groups (A, B, C). *** *p* < 0.001: significantly higher than experimental group C (Tukey post hoc test). *** p* = 0.036: significantly lower than experimental group A (Tukey post hoc test).

**Table 1 biology-12-00859-t001:** Experimental group set-up.

Experimental Group	Nutritional Stress	No. of Ind./Bowl	No. of Replicates
A Controls	Absent—feeding supply (100%—*ad libitum*)	2 (1 Yy—focal + 1 yy)	24
B	Intense—feeding supply (25%)	2 (1 Yy—focal + 1 yy)	24
C	Extreme—feeding supply (0%)	2 (1 Yy—focal + 1 yy)	24

## Data Availability

The analytical data will be made available to interested parties upon request.

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
