# Peer review of "Influence of Nutritional Stress on Female Allocation and Somatic Growth in the Simultaneously Hermaphroditic Polychaete Worm Ophryotrocha diadema"

_biology, 2023, doi:10.3390/biology12060859_

Round 1

Reviewer 1 Report

The words in title must be not capitalized. ...

Please revise reference number 18 and 19 that were not related to the text please check the citation of all presented references. ..last lines in the introduction sections are very bad presented need to be rephrased again  ..... in materials. ..what the authors based on in the classifications of groups...in addtionthe writing if this part is very bad please check ..where is study limitations???

Where is the ethical commitee approval number?

Deleted some references from (21'22'23 24 25 26) much more refrence present ...figure 1 of very bad resolution and quality. ...in all tables mean and standerd deviation must be together in one column 

Conclusion must be rewritten 

The quality of English is very bad need to be rephrased 

Author Response

Reply to Reviewer 1

The words in title must be not capitalized.

R: we corrected title as suggested

Please revise reference number 18 and 19 that were not related to the text please check the citation of all presented references.

R: we revised the citations of all presented references, as suggested

Introduction

last lines in the introduction sections are very bad presented need to be rephrased again.

R.: We substituted the paragraph: “Furthermore, to verify an eventual influence of nutritional stress also on the somatic growth of O. diadema individuals, the body size was measured, i.e., number of initial (at the beginning of the experiment) and final (at the end of the experiment) setigerous segments estimated.”

With the following new one:

“Moreover, to verify the influence of nutritional stress on the somatic growth rate of focals, the body size of individuals was estimated. In particular, we measured the number of setigerous segments at the beginning and at the end of the experimental period, whereas the body growth rate was estimated as difference between the final and the initial number of these segments.”

Materials and Methods

in materials. what the authors based on in the classifications of groups.

R.: Since in the present experiment we tested effects of nutritional stress, experimental groups were classified on the three different types of nutritional stress: absent, intense, and extreme, maintaining the rest of all conditions constant. Similar experimental design was set up also by Locher and Baur, 2002.

in addition, the writing of this part is very bad please check

R.: Our manuscript underwent extensive English revisions by our English revisor (mother language).

where is study limitations???

R.: Food deprivation could be variable, occasional, but sometime frequent in natural environments, due to different environmental conditions. This represents a limitation of our study because we can only simulate such deprivation in laboratory, since to perform all possible conditions such as those that can occur in nature is difficult. We included this consideration in conclusion session.

Where is the ethical committee approval number?

R.: As far as we know, in Italy ethics committee approval is not required for invertebrate animal testing.

Deleted some references from (21'22'23 24 25 26) much more reference present ...

R.: We eliminated the references 21-22 and, consequently, we renumbered the references session.

Figure

Figure 1 of very bad resolution and quality. ...

R.: In the revised version, we eliminated the Figure 1.

Tables

in all tables mean and standard deviation must be together in one column 

R.: We eliminated all the tables (except the table 1) as suggested by the other referees. Graphs are present in the text.

Conclusions

Conclusion must be rewritten 

R.: We deeply modified conclusions, as suggested.

Comments on the Quality of English Language

The quality of English is very bad need to be rephrased 

R.: As mentioned above, our manuscript underwent extensive English revisions by our English revisor.

Reviewer 2 Report

The authors of the manuscript (Biology-2395805) demonstrate a progressive decline in female allocation as well as somatic growth due to nutritional stress in O. diadema species. They propose that climate change could trigger a lack of nutrients in the marine environment, which in turn will have a negative impact on the reproductive capacity of the species, altering the entire ecosystem.

The experimental design and the subsequent statistical analysis are adequate for the aims of the study and the results, discussion, and conclusions are exposed efficiently.

However, I consider that the manuscript is not suitable for publication in its present form. It needs to be revised before a decision can be made on its suitability for publication. I have some suggestions and questions regarding the results obtained:

1.   Rewrite the paragraph of the summary:

“ Since our results revealed a progressive decrease in female allocation and somatic growth of O. diadema individuals with increase of nutritional stress, our results clearly demonstrate a significant influence of nutri-tional stress on both, female allocation in terms of number of cocoons and number of eggs, as well as on body growth rate”.

The words "our results" appear twice and the main idea is repeated.

2.   Change the form of bibliographic citations:

“These predictions are already supported by few empirical tests [2, 3, 4, 5, 6, 7, 8, 9, 10, 11,12, 13]”.    Change to [2-13]. See introduction

Ophryotrocha diadema, a 1-mm long marine polychaete worm, is a simultaneous her-maphrodite with external fertilization (Figure 1). All data on life cycle [20] and mating system [21, 22,23, 24, 25, 26] have been obtained through laboratory observations. O. di-adema populations live among the organic sediments of fouling fauna of Californian har-bours and belong to marine zooplankton species. Change to [21-26].

3.    Figure 1 should be a bar scale (mm).

4.    Graphy 1, 2, and 3 should be named as Figures 2, 3 and 4 in the text.

5.    The differences between groups A, B and C for the variables analyzed should be indicated in the graph itself (with letters or symbols). Thus, tables 3, 4, 5, 6 and 7 would not be necessary, since they do not provide more information than what is already indicated in the text and in the graph.

6.    Why have only Yy individuals been analyzed?

7.    The authors took samples twice a week for 21 days. However, the results are only given at the end of the study. Have the authors analyzed temporal variation? How did the data for each variable vary as a function of time?

Author Response

Reply to Reviewer 2

  1. Rewrite the paragraph of the summary:

“Since our results revealed a progressive decrease in female allocation and somatic growth of O. diadema individuals with increase of nutritional stress, our results clearly demonstrate a significant influence of nutri-tional stress on both, female allocation in terms of number of cocoons and number of eggs, as well as on body growth rate”.

The words "our results" appear twice and the main idea is repeated.

R.: We substituted the above mentioned paragraph, with the follow: “Our results revealed a progressive decrease in female allocation, in terms of number of cocoons and number of eggs, and body growth rate of O. diadema individuals with increase of nutritional stress”.

  1. Change the form of bibliographic citations:

“These predictions are already supported by few empirical tests [2, 3, 4, 5, 6, 7, 8, 9, 10, 11,12, 13]”.    Change to [2-13]. See introduction

Ophryotrocha diadema, a 1-mm long marine polychaete worm, is a simultaneous her-maphrodite with external fertilization (Figure 1). All data on life cycle [20] and mating system [21, 22,23, 24, 25, 26] have been obtained through laboratory observations. O. diadema populations live among the organic sediments of fouling fauna of Californian harbours and belong to marine zooplankton species. Change to [21-26].

R.: We change the form of bibliographic citations, as suggested.

  1. Figure 1 should be a bar scale (mm).

R.: In the revised version, we eliminated the Figure 1, as suggested by the other referees.

  1. Graphs 1, 2, and 3 should be named as Figures 2, 3 and 4 in the text.

R.: Done. However, we decided to eliminate Figure 1 and, consequently, Graphs 1,2, and 3 were named as Figures 1,2, and 3.

  1. The differences between groups A, B and C for the variables analyzed should be indicated in the graph itself (with letters or symbols). Thus, tables 3, 4, 5, 6 and 7 would not be necessary, since they do not provide more information than what is already indicated in the text and in the graph.

R.: We eliminated all the tables (except the table 1), as suggested. “Graphs” were substitute with Figures” that are present in the text. We highlighted the differences between groups by asterisk.

  1. Why have only Yy individuals been analyzed?

R.: In order to estimate our parameters at the INDIVIDUAL level, such parameters were measured only in focal individuals (O. diadema Yy with yellow phenotype).

To this aim, in all the experimental groups, we set up 2 individuals per bowl (1 Yy individual with yellow phenotype and 1 yy with white phenotype). Consequently, we collected data related only to the focal individuals estimating the number of yellow cocoons and the number of (yellow) eggs/cocoon as well as the number of initial and final setigerous segments of yellow individuals in order to estimate their body growth rate.

In this way all the parameters have been estimated at the individual level.

There are not differences in the reproductive behaviuor, reproductive parameters and life-history traits between O. diadema Yy individuals and O. diadema yy individuals (Sella, 1985). Therefore, if we would estimate all the parameters also in O. diadema yy, it would be just a duplication of data.

In the experiment, the presence of O. diadema yy with white phenotype was essential to promote egg trading reproductive strategy, since isolated O. diadema individuals, despite being simultaneous hermaphrodites, never spawn cocoons.

If we had set up 2 yellow individuals, or 2 white individuals per bowl, all the data collected would have resulted per pair and not per individual.

  1. The authors took samples twice a week for 21 days. However, the results are only given at the end of the study. Have the authors analyzed temporal variation? How did the data for each variable vary as a function of time?

R.: The goal of the present experiment was to estimate the total allocation of female reproductive resources and the growth rate in the three different nutritional regimes. Thus, we did not consider temporal variation of the female allocation and the growth rate, but only final results of such nutritional stress, estimating the total number of cocoons, the total number of eggs/ cocoon and the growth rate as difference between the final and the initial number of setigerous segments.

However, regarding to temporal variation, in the first group, where the nutritional stress was absent, the female allocation and the body growth rate was rather constant. In the second experimental group, where the nutritional stress was intense, the female allocation and the body growth rate was decreasing. Finally, in the third experimental group, where the nutritional stress was extreme, all the parameters estimated were deeply decreasing.

Reviewer 3 Report

12th line of the abstract: I'm not sure it is really too important to make any mention of climate change - its another subject entirely how global warming will affect phytoplankton, I'd say this subject is too unrelated to the current study to even be worth a mention in this manuscript. This study is just a small-scale laboratory study so I do not think its too useful to use it to make speculation related to global climate change.

43rd line on page 2: The phrase "number of cocoons laid and number of eggs/ cocoon estimated" is confusing - please clarify exactly what is meant here.

last paragraph page 4: Its good that the authors checked for data normality, but it would be good if they could also check for homogeneity of variances, using e.g. Levene's test or Cochran's test. If the assumption for the ANOVA of homogeneity of variances is not met, then measures may need to be taken such as data transformation.

Table 2 and 5: These tables just repeat information shown on the graphs and written in the main text, so I do not think they are needed.

Table 6: this is exactly the same information as in the main text directly above it - please delete this table.

Last paragraph before the discussion: again here there are specific details/numbers with unnecessary repetition in the tables and in the main text. Here I would say that the numbers in the main text in this paragraph can probably be deleted and the information just presented in the table.

Conclusions paragraph: As I said before, I do not think there is any need in this manuscript to bring up concepts of global warming. I would say that this whole paragraph can be deleted.

2nd line on page 2: change to "species such as", and please also make a similar change on the next line.

last paragraph on page 3: please italicise the species names in this paragraph.

22nd line page 4: change to "which lasted 21 days"

Graph 1 and 2: please italicise the species name in the graph caption.

Last sentence on page 8: This sentence here is really long and difficult to understand, please clarify.

9th line page 9: change to "when a high number".

27th line page 9: delete the 2nd "of" on this line.

34th line page 9: is it meant to be "focal individuals" here?

Author Response

Reply to Reviewer 3

12th line of the abstract: I'm not sure it is really too important to make any mention of climate change - its another subject entirely how global warming will affect phytoplankton, I'd say this subject is too unrelated to the current study to even be worth a mention in this manuscript. This study is just a small-scale laboratory study, so I do not think its too useful to use it to make speculation related to global climate change.

R.: We eliminated from the Abstract all the part related to the global warming, as suggested.

43rd line on page 2: The phrase "number of cocoons laid and number of eggs/ cocoon estimated" is confusing - please clarify exactly what is meant here.                                                                        

R.: "number of cocoons laid and number of eggs/cocoon estimated" means that we estimated the number of cocoons laid and also the number of eggs inside these cocoons (= eggs/ cocoon).

However, since the phrase is misleading, we modified the phrase: “Since previous studies showed that the number of sperm is not influenced by social and environmental conditions [2, 4, 12, 13], influence of nutritional stress was estimated only on female allocation (i.e., number of cocoons laid and number of eggs/ cocoon estimated).”

With the following one:

“Since previous studies showed that the number of sperm is not influenced by social and environmental conditions [2, 4, 12, 13], influence of nutritional stress was estimated only on female allocation (i.e., number of cocoons laid and number of eggs/ cocoon).”

last paragraph page 4: Its good that the authors checked for data normality, but it would be good if they could also check for homogeneity of variances, using e.g. Levene's test or Cochran's test. If the assumption for the ANOVA of homogeneity of variances is not met, then measures may need to be taken such as data transformation.

R.: Levene test was added to the Statistical analyses and the phrase: “Distributions of cocoon numbers, eggs/individual and the growth rate were tested for normality using the Kolmogorov–Smirnov one-sample test (P > 0.05).” was modified as follows:

“Distributions of cocoon numbers, eggs/individual and the growth rate were tested for normality using the Kolmogorov–Smirnov one-sample test and for homogeneity of variances using Levene’s test (P>0.05).”

Table 2 and 5: These tables just repeat information shown on the graphs and written in the main text, so I do not think they are needed.

R.: We eliminated all the tables from the text (except the table 1), as suggested also by the other referees.

Table 6: this is exactly the same information as in the main text directly above it - please delete this table.

R.: As mentioned above, we eliminated all the tables (except the table 1).

Last paragraph before the discussion: again, here there are specific details/numbers with unnecessary repetition in the tables and in the main text. Here I would say that the numbers in the main text in this paragraph can probably be deleted and the information just presented in the table.

R.: Since we preferred to maintain the same trend in the presentation of the data, we eliminated the table (table 7) from the text, maintaining the data in the main text.

Conclusions paragraph: As I said before, I do not think there is any need in this manuscript to bring up concepts of global warming. I would say that this whole paragraph can be deleted.

R.: We deeply modified the conclusion paragraph as follows:

Our study supports the importance of the availability of food resources in O. diadema. Food deprivation could be variable, occasional, but sometimes frequent in natural environments, due to different environmental conditions. This represents a limitation of our study because we can only simulate this deprivation in the laboratory, since performing such an experiment in all possible conditions such as those that can occur in nature is difficult. It is known that abiotic factors, such as increased temperature and acidification, have a significant impact on the biomass of phytoplankton and zooplankton communities exposed, significantly affect their survival, behaviour, reproduction and, in general, their overall population dynamic. For example, evidences suggest that marine zooplankton generally experience food levels above subsistence values, caused by the reduction in the phytoplankton biomass richness associated to the increase of marine water temperature [28, 29]. Similarly, a reduction in trophic richness could have a strong impact on the reproductive output and, thus, also on the population growth rate of this marine polychaete worm. Considering that zooplankton represents the main trophic connection between primary producers (phytoplankton) and the final consumers, any biotic or abiotic factor that interferes with their population dynamics could lead to repercussions at the level of the food chain and biodiversity.

Comments on the Quality of English Language

2nd line on page 2: change to "species such as", and please also make a similar change on the next line.

R.: Done

last paragraph on page 3: please italicise the species names in this paragraph.

R.: Done

22nd line page 4: change to "which lasted 21 days"

R.: Done

Graph 1 and 2: please italicise the species name in the graph caption.

R.: All graphs were transformed in figures, as suggested by other reviewer. In all graphs we italicized the species name, as suggested. However, the SPSS software does not consent to italicized only a little part of the graph caption, thus we decided to italicized all graph captions.

Last sentence on page 8: This sentence here is really long and difficult to understand, please clarify.

R.: We modified the sentence

“Thus, such results suggest the presence of other factors, for example life history traits of species (i.e., mating behaviour as mate guarding, mate searching, courtship, fertilization rate; food competition, food searching etc or various environmental conditions and stressors (i.e., physical obstruction, temperature shock, physical damage, food deprivation etc) involved in the overall energy budget and consequently also in the allocation of reproductive resources and therefore in trade-off between sexual functions.”

with the follow one:

Several factors are involved in the overall energy budget of hermaphrodites. Such factors could be life history traits of species (i.e., mating behaviour as mate guarding, mate searching, courtship, fertilization rate; food competition, food searching etc) or various environmental conditions and stressors (i.e., physical obstruction, temperature shock, physical damage, food deprivation etc). Thus, in hermaphrodites, all these factors could influence the allocation of reproductive resources and consequently also trade-off between sexual functions”.

9th line page 9: change to "when a high number".

R.: Done

27th line page 9: delete the 2nd "of" on this line.

R.: Done

34th line page 9: is it meant to be "focal individuals" here?

R.: Yes. We corrected it.

Reviewer 4 Report

I found this simple experimental study and the relative manuscript weak in some parts. Several revisions are needed in my opinion to improve its value.

Please double-check the English style of the entire document to improve its clarity and fluency.

Moreover, re-organize the manuscript following the style of the Journal (Abstract length, sections, references)

Regarding contents, please avoid keywords already reported in the Title, Abstract section is wordy and not so effective, try to focus more on the critical results of the study.

The study model features should be moved from the material and methods to the introduction section. Moreover, this section should be arguments about the environmental relapses of this study. Is food deprivation at these tested levels frequent in natural environments? Only some scarce support in this regard was related to climate change, but from an environmental point of view, this is so limited, variable and occasional.

In the experimental setup, more parameters should be considered and reported, such as dissolved oxygen or temperature during all the experimental phases.

Double-check all the scientific names and put them in italics in the entire document (captions, references, etc).

The discussion lacks a real comparison between the data obtained in this study and previous experimental studies or environmental reports. Please better argue the last part of this section considering all the possible points of view. 

Also in the conclusion section, similarly to the introduction, a real link between global warming/climate change and food deprivation was missed. Please reconsider your results based on current environmental conditions.

Best regards 

The Reviewer

Extensive editing of the English language is required

Author Response

Reply to Reviewer 4

Please double-check the English style of the entire document to improve its clarity and fluency.

R.: Our manuscript underwent extensive English revisions by our English revisor (mother language).

Moreover, re-organize the manuscript following the style of the Journal (Abstract length, sections, references

R.: We re-organize the manuscript, as suggested. Simple Summary was added, we checked the sections and references.

Simple Summary:

Simultaneous hermaphrodites adjust their reproductive resources mainly according to the mating opportunities. However, several other factors are involved in the overall energy budget of hermaphrodites. Such factors could be life history traits of species or various environmental conditions and stressors. Thus, in hermaphrodites, all these factors could influence the allocation of reproductive resources and consequently also trade-off between sexual functions. In the present experiment we investigated the trade-off between food deprivation and the investment of resources to the female allocation and somatic growth in the simultaneously hermaphroditic polychaete worm Ophryotrocha diadema, exposing focal individuals to the three different nutritional regimes. Our results clearly demonstrate a significant influence of nutritional stress on both, female allocation and on body growth rate.

Regarding contents, please avoid keywords already reported in the Title.

R.: Done

Abstract section is wordy and not so effective, try to focus more on the critical results of the study.

  1. We modified Abstract section, as suggested.

The study model features should be moved from the material and methods to the introduction section.

R.: We moved the study model features from the material and methods to the introduction, as suggested.

Moreover, this section should be arguments about the environmental relapses of this study.

Is food deprivation at these tested levels frequent in natural environments? Only some scarce support in this regard was related to climate change, but from an environmental point of view, this is so limited, variable, and occasional.

R.: We discussed about the environmental relapses of this study in the Conclusion section that was deeply modified.

In the experimental setup, more parameters should be considered and reported, such as dissolved oxygen or temperature during all the experimental phases.

R.: During the experimental period we maintained constant temperature, set up at 20̊°C, as written in the text. We have not measured dissolved oxygen, but since we set up 2 adult individuals per bowl in all the experimental groups, we suppose that the oxygen consumption was constant. We added these details to the Experimental set up section, as suggested.

Double-check all the scientific names and put them in italics in the entire document (captions, references, etc).

R.: Done

The discussion lacks a real comparison between the data obtained in this study and previous experimental studies or environmental reports. Please better argue the last part of this section considering all the possible points of view. 

R.: As far as we know, there are only two similar previous experimental studies related to the environmental stressors (Hughes et al., 2003) and food deprivation (Locher and Baur, 2002) in the simultaneous hermaphrodites that we argued in details in the new version of the discussion section.

Also in the conclusion section, similarly to the introduction, a real link between global warming/climate change and food deprivation was missed. Please reconsider your results based on current environmental conditions.

R.: We deeply modified the conclusions’ section.

Reviewer 5 Report

The paper entitled “ Influence of Nutritional Stress on Female Allocation and So-matic Growth in the Simultaneously Hermaphroditic Poly-chaete Worm Ophryotrocha diadema” by Dáša Schleicherová, Marino Prearo and Alfredo Santovito contains some information od influence of amount of food on female allocation and somatic growth of Ophryotrocha diadema. In general, the work is interesting, although it contains many errors, which I indicate in pdf file.  

General comment:

1.      Why O. diadema were fed only aqueous suspension of minced parboiled spinach? Maybe it would be better to feed them other food?

2.       Some graphs in paper are unnecessary.

Author Response

Reply to Reviewer 5

The paper entitled “Influence of Nutritional Stress on Female Allocation and Somatic Growth in the Simultaneously Hermaphroditic Poly-chaete Worm Ophryotrocha diadema” by Dáša Schleicherová, Marino Prearo and Alfredo Santovito contains some information od influence of amount of food on female allocation and somatic growth of Ophryotrocha diadema. In general, the work is interesting, although it contains many errors, which I indicate in pdf file.

R.: The authors thank the referee for his positive comment on our paper.

General comment:

  1. Why O. diadema were fed only aqueous suspension of minced parboiled spinach? Maybe it would be better to feed them other food?

R.: We rear O. diadema populations for more than 30 years in the laboratory feeding them with aqueous suspension of minced parboiled spinach. Despite they are not fed with their natural food found in the wild (green algae), they are perfectly adapted to spinach. We tried to feed them with different food, mainly with vegetables, as minced carrots, or tomatoes etc., but they died in 24h. It is possible to feed them also with dry fish food. However, the fish food does not contain the lutein. The lutein is present in the spinach, and it is responsible of yellow phenotype of eggs in O. diadema Yy. O. diadema Yy are able to absorb lutein from spinach. Instead, O. diadema yy, with white phenotype are not able to absorb it. In this way, if fed with spinach, we could have 2 strains of the same species: O. diadema Yy and O. diadema yy. For this reason, we prefer to feed them with aqueous suspension of minced parboiled spinach.

  1. Some graphs in paper are unnecessary.

R.: We eliminated all the tables from the text (except the table 1), as suggested by other referees.

R.: We checked the pdf file and we noted that many errors are related to the grammar, English and punctuation. However, our manuscript underwent extensive English revisions by our English revisor (mother language) and now it is improved.

Moreover, we deeply modified Conclusion section, as suggested.

Round 2

Reviewer 1 Report

All comments were addressed by authors  ut please revise the citations well and exclude  any self citations or old references...thanks now I can recommend publication of the manuscript 

Fine

Reviewer 4 Report

Dear Authors,

thank you for seriously considering my previous comments on your manuscript, which now appears better organized and written. Moreover, the soundness of the document was enhanced considering more relapses and aspects.

Best regards